

# The conflict between sampling resolution and stratigraphic constraints from a Bayesian perspective: OSL and radiocarbon case studies

Guillaume Guérin[1], Pierre Guitton-Boussion[1], Imène Bouafia[2], Anne Philippe[2]

[1] Univ Rennes, CNRS, Géosciences Rennes, UMR 6118, 35000 Rennes, France
[2] Jean Leray Laboratory of Mathematics (LMJL), UMR6629 CNRS - Université de Nantes, France

*Correspondence to*: Guillaume Guérin (guillaume.guerin@univ-rennes.fr)

**Abstract.** Bayesian modelling is often implemented in geochronology and its applications to geomorphology, archaeology,
etc. The rationale behind such practices is the aim to improve robustness, precision and accuracy thanks to the use of prior knowledge regarding the studied sites, and in particular the order of samples constrained by stratigraphy. All chronological models tested in this study (OxCal, Chronomodel and BayLum) use the same mathematical model to handle stratigraphic constraints. However, this model has been shown to lead to estimation biases. First, this bias is illustrated with BayLum modelling on a high-resolution OSL dataset. Then, this paper compares statistical inferences obtained with the three above-
mentioned modelling software on the Neolithic East mound of Çatalhöyük (Turkey). For this site, 49 radiocarbon ages were obtained with the aim to determine the start of occupations at this locality. Interestingly, age uncertainties are rather large, because of calibration curve plateaus. Therefore, the conditions for estimation biases are met. We discuss the behaviour of the different models and show that caution must be taken when modelling results are at odds with measurements. While OxCal, Chronomodel and BayLum are all affected by a spread in ages resulting from their common model of stratigraphic errors,
Chronomodel suffers from a great loss of precision and OxCal, through the phase model, concentrates ages undesirably. We also conclude that the onset of occupations at Çatalhöyük was probably earlier than previously thought based on the OxCal model.

## 1. Introduction

In geochronology, Bayesian modelling is often used to include prior knowledge for estimating time sequences. Perhaps the
most important, and most commonly used prior knowledge, is the order in which events are placed in time, through the principle of stratigraphy. Typically, stratigraphic sequences comprising several sedimentary layers are dated; if Layer a was deposited prior to Layer b, then the deposition age of Layer a ($t_a$) is greater than that of Layer b ($t_b$). While this statement is straightforward, its transcription in a mathematical model is the focus of the present article.

To our knowledge, all chronological models use the same prior probability on the dates:

$$\pi(t_a, t_b) = \begin{cases} const\ if\ t_a > t_b, \\ 0\ otherwise \end{cases}, \tag{Eq. 1}$$





Such prior is indeed implemented in OxCal (Ramsey, 2009), Chronomodel (Lanos and Philippe, 2017; Lanos and Dufresne, 2019) and BayLum (Combès and Philippe, 2017; Christophe et al., 2020; Philippe et al., 2019; Guérin et al., 2021). In addition, in OxCal by default the prior density of the time span between dates is chosen to be uniform, following Nicholls and Jones (2001), i.e., a priori all durations between the first and the last date are given equal probability. These three software

programmes address specific needs: OxCal was initially developed for radiocarbon (Hajdas et al., 2021), while Chronomodel was designed to combine various dating methods; finally, BayLum was specifically conceived for Optically Stimulated Luminescence (OSL; Murray et al., 2021) dating (note: it also allows radiocarbon calibration and modelling, like OxCal accommodates non-radiocarbon ages in the form of gaussian probability densities).

The ordering constraint mentioned above may then be extended to series of $n$ ages: the prior probability is constant when the

sample ages are in order and null otherwise. This choice of prior was the subject of a study by Steier and Rom (2000), who concluded that Equation 1 may lead to estimation biases in the following cases:

1) when outliers are present, this probability distribution is not robust with respect to their presence, causing a shift in all ages; however, the present article does not focus on outliers;

2) when, for a fixed study period, the number of measured samples is increased. In such a case, in particular the ages of extreme

(youngest and oldest) samples are shifted away from each other, and from the true value;

3) when the true age discrepancy between successive samples is too small compared to measurement uncertainties. In such a case, the age of the older event tends to be overestimated, and that of the younger event underestimated.

Further, Steier and Rom (2000) stated that the outcome of models using Eq. (1) may well produce artificially low uncertainties and thus increase the impression of precision (because the highest posterior density intervals are reduced), but at the cost of

reduced accuracy: in their simulated test cases, the true age value often lies outside the 95% credible interval. Such a behaviour is not wanted and it is the reason why Nicholls and Jones (2001) proposed a uniform prior on phase duration. In practice, the two problems 2) and 3) listed above (excluding the question of outliers) arise when the sampling resolution is too high, i.e., when the true age difference between successive samples is small compared to the measured age difference. It should be noted here that the latter comparison, between true and measured ages, only makes sense when considering measurement

uncertainties: problems arise when probability densities of measured ages overlap. From this, we may deduce that issues will arise more frequently with imprecise measurements, and, thus, should occur more with OSL than with radiocarbon, simply because OSL is less precise than radiocarbon. Note: OSL and radiocarbon being the two most widely used dating methods for the study of the past 50,000 to 100,000 years, we will essentially focus on these methods in this article; nevertheless, all dating methods must be concerned by the problem.

The loss of accuracy arises because the modelled ages are spread by the model (Eq. 1), i.e., the age of younger samples becomes underestimated while the age of the older samples gets overestimated by the strict ordering constraint. In the following, we refer to this observation as the 'spread effect'. Visually, the spread effect is recognised by comparing measured probability densities with posterior probability densities (i.e., the ages estimated with the model): a clear pattern is that, for a pair of samples yielding indistinguishable measured ages, the model artificially creates two distinct posterior densities. In such cases,



the model excludes the solutions in which the two samples have a negligible true age difference. As a result of this effect, the development of dating methods and in particular the general increase in the number of dated samples per study (e.g., in high-resolution chronologies published by Stevens et al., 2018) makes this estimation problem more and more acute.

In reply to Steier and Rom (2000), Ramsey (2000) pointed to boundaries as a mean to overcome the spread effect. Indeed, in OxCal dated samples are generally modelled between two undated events: the start of the phase including the dated samples

and its end. By default, in OxCal the uniform phase model (see for example Naylor and Smith, 1988) is implemented. It should be emphasized here that in the present article, we only consider this uniform phase model, in which all ages have a uniform probability density between the phase boundaries. Inside such a phase, the prior probability of ages is uniform conditionally to the phase boundaries. If $t_\alpha$ is the age of the end of a phase, and $t_\beta$ the age of its beginning, then the prior becomes:

$$\pi(t_\alpha, t_i, t_\beta) \propto \frac{\Pi_i \pi((t_\alpha, t_i, t_\beta))}{(t_\beta - t_\alpha)^n}, \text{(Eq. 2)},$$

where $t_i$ denotes a suite of $n$ ages. Such a model has been shown to lead to a 'concentration effect' (Lanos and Philippe, 2018): the greater is $n$, the greater is the probability that $t_\beta - t_\alpha$ is small in the model outcome (i.e., lower differences between $t_\beta$ and $t_\alpha$ are favoured by the model). In other words, the phase model tends to cluster all ages inside a phase: $t_\alpha \approx t_i \approx t_\beta$. Another effect of the phase model is that phase boundaries ($t_\beta$ and $t_\alpha$) may extend well outside the range of measured $t_i$ values, all the more when the number of ages inside a phase is small (e.g., Guérin et al., 2023).

To summarise, Ramsey (2000) argued that the spread effect induced by Eq. (1) is compensated in OxCal by Eq. (2), i.e. that the spread effect arising from imposing stratigraphic constraints is countered by the concentration effect arising from the phase model. The first purpose of this article is to illustrate the issue at stake using various datasets obtained from different methods (namely, OSL and radiocarbon), and to compare different Bayesian chronological models: OxCal, Chronomodel and BayLum. By comparison with OxCal, in Chronomodel the event model of Lanos and Philippe (2018) is the basis for chronological

modelling; every date or age is indeed encapsulated in one event – and one event may contain several dates or ages (for example, speleothems or corals may be jointly dated by U/Th and radiocarbon). Thus, in Chronomodel the concentration effect should not appear; however, the event model introduces an additional time uncertainty, of a similar magnitude as the measured age uncertainty. Finally, BayLum is a somewhat simpler model regarding the modelling of ages in stratigraphic constraints, in the sense that only the spread effect should be visible and that no additional uncertainty is added to measured ages. To

summarise, OxCal is affected by the spread and concentration effects, Chronomodel by the spread effect and by additional uncertainties, and BayLum by the spread effect only.

The purpose of this paper is to illustrate the issue that may be encountered in chronological modelling of OSL and ${}^{14}$C-based chronologies. We show how it can be recognised upon scrutiny by comparing ages modelled with or without stratigraphic constraints. To this end, we use two real datasets rather than modelled ones to emphasize the possibility to detect modelling

artefacts without knowing the true age of each sample. First, an already published, high-resolution OSL dataset, is modelled using BayLum. Issues related to increased sampling resolution are demonstrated using the open-source ArchaeoPhases R package (Philippe and Vibet, 2020). To compare chronological models when confronted with high-resolution datasets, we turn





to radiocarbon dating. Indeed, in BayLum OSL measurements are combined in a hierarchical model linking regenerative doses, individual equivalent dose estimation, the central dose parameter of interest, etc. Conversely, OSL ages can be included in

OxCal and in Chronomodel, but only in the form of Gaussian probability densities (in practice, an age and its uncertainty). Therefore, to make the model comparison as straightforward as possible, we use a high-resolution radiocarbon dataset. Each of the three tested models indeed performs calibration of measured $^{14}$C ages, using similar Markov Chain Monte Carlo (MCMC) algorithms.

## 2.   Materials and methods

### 2.1. OSL dating and BayLum

Loess plateaus often offer long sedimentary sequences spanning long periods of time; as such, they provide valuable records of sediment accumulation together with climatic and environmental proxies. During periods of fast sediment accumulation, series of indistinguishable OSL ages may be obtained over several meters of depth. Such is the case of the Jingbian loess plateau, recently studied by Stevens et al. (2018). For our study, we selected of cluster of 10 samples deposited around 15 ka

ago (Fig. 1); 7 of these 10 samples are consistent with the average age (14.6 ka) at one standard deviation and 3 at two standard deviations. In other words, all ages are consistent with a short, fast deposition event (short and fast compared to individual age uncertainties). Yet, they correspond to 90 cm of loess accumulation; therefore, their true ages are ordered following the stratigraphy. To investigate the issue posed by Eq. (1) in the modelling of stratigraphic constraints, we used BayLum to model the stratigraphic sequence. To take systematic errors into account (which is important when imposing the stratigraphic

constraints, because some errors are not random), we built a Theta matrix following Guérin et al. (2021); following the assumptions made by the authors of the original study, we considered the following sources of systematic errors:

- calibration of gamma spectrometers for the concentrations in K, U and Th (1.5% uncertainty on each value);

- calibration of the laboratory-source dose rates (3% uncertainty);

- internal dose rate (0.01 Gy.ka$^{-1}$, absolute uncertainty);

- water content, taken to be $15 \pm 5$ %; we treated all the uncertainty associated to this parameter as corresponding to a systematic error.



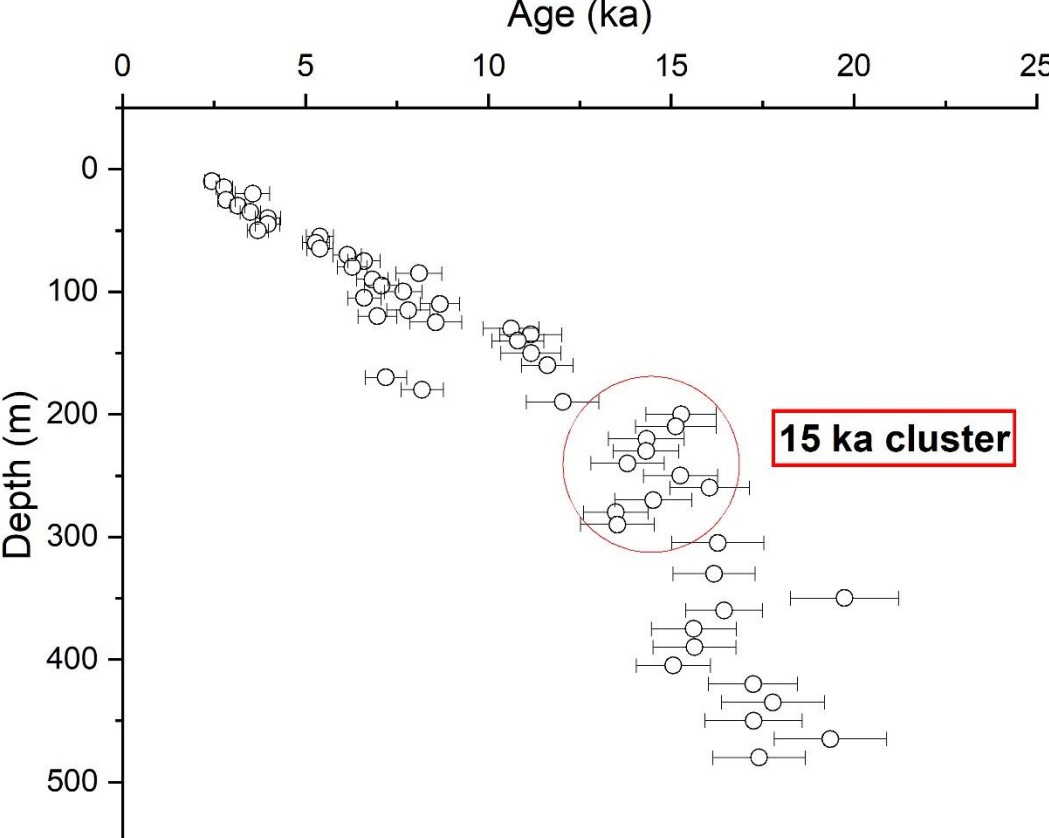

**Figure 1. OSL ages as published by Stevens et al. (2018) for the study of a loess Plateau in Jingbian, China. The subset**
**of 10 samples analysed in the present study lies inside the red circle. In this figure, error bars correspond to one sigma,**
**i.e., to the 68% confidence interval.**

As pointed out in the introduction (see issue number 2), for a fixed duration of events the number of dated samples is a key
parameter in the issue at stake: the more samples there are, the wider is the resulting spread in ages modelled in order.
Therefore, we investigated the effect of including various numbers of samples on the estimated duration between the deposition
of the first and last samples. To quantify the effect, we used the open-source R package ArchaeoPhases to estimate this
duration. We also looked at the joint probability between ages estimated with and without imposing stratigraphic constraints,
to estimate the consistency between the model output and the input data, as a function of the number of samples included in
the model.





### 2.2. Comparison of OxCal, BayLum and Chronomodel on a radiocarbon dataset


For the purpose of model comparison, we selected a high-resolution radiocarbon dataset, with large uncertainties. In general, measurements of the $^{14}$C/$^{12}$C ratios display <1% uncertainty, so we decided to focus on periods affected by plateaus in the calibration curve (Reimer et al., 2020) – i.e., periods for which similar $^{14}$C/$^{12}$C ratios correspond to various true ages. One study that appears to correspond to these criteria is the chronological investigation led by Bayliss et al. (2015) at the famous

archaeological site of Çatalhöyük, in Turkey. 49 radiocarbon ages from 46 samples (3 measurements were replicates and combined with their counterpart) were obtained from the stratigraphy, spanning a period from 7,600 BC until 6,600 BC. Table S1 provides all selected radiocarbon ages. If ages were distributed homogeneously, ~20 years would separate two successive samples. This order of magnitude for the true age difference is much smaller than typical age uncertainties: analytical uncertainties on the $^{14}$C/$^{12}$C ratio are ~40-50 years, and after calibration these uncertainties are largely increased.

The purpose of the present study being to compare OxCal, Chronomodel and BayLum on the same dataset, we will not discuss here data selection and modelling choices made by Bayliss et al. (2015). Thus, we reproduced their model, based on their discussion but also their modelling choices. Indeed, OxCal offers the possibility to define phases, encapsulated phases (i.e., phases inside other, larger phases), hiatuses, terminus ante- and post-quem, boundaries, etc. For example, one may wonder why boundaries are defined in the model sometimes before, in other places between, and/or after phases. But this is not the

purpose of our work here, so we take the arguments of Bayliss et al. (2015) at face value and stick to their preferred model.

By comparison with OxCal, it is important to note that Chronomodel and BayLum offer fewer options to model chronological sequences: in the latter two modelling programmes, events (in Chronomodel) and ages (in BayLum) are stratigraphically constrained according to the Harris matrix provided by Bayliss et al. (2015; their Fig. 3), but are not affected by the definition – or absence of definition – of boundaries and phases. This lack of modelling choices actually appears to be a positive feature

in our view, in the sense that it tends to provide less arbitrary results. One clear example of issues posed by the phase model can be seen, *e.g.,* in the work of McPherron et al. (2012), who modelled 10 radiocarbon samples from one unique layer. In their study, defining a phase – i.e., saying that there was a 'before' and an 'after' – affects the chronological inference by (i) concentrating the ages and (ii) excluding ages considered as outliers. In our view, there can be no outlier detection when no stratigraphic inversion is detected; as discussed in detail in Guérin et al. (2023), from first principles the start of a chronological

phase should not possibly be defined after the oldest sample and before the youngest sample.

Fig. 2 presents the 95% credible interval (C.I.) of the ages calibrated with OxCal, Chronomodel and Baylum. For calibration, IntCal20 was used in OxCal and BayLum, while it is not available in the current version of Chronomodel. Hence, IntCal13 was used in Chronomodel. Most age intervals are very similar and most differences occur between Chronomodel on the one hand, BayLum and oxCal on the other. We attribute these differences to the change in the calibration curve. It should be noted

here that Fig. 2 only includes the ages that are not considered as outliers in subsequent modelling (and in Bayliss et al., 2015); indeed, including here ages that are clearly off the stratigraphic record would render this figure difficult to read. However, all ages are given in Table S1.



**Figure 2. Radiocarbon ages from Çatalhöyük calibrated with OxCal, Chronomodel and BayLum. Note: IntCal20 was used in OxCal and BayLum, while it is not available in the current version of Chronomodel. Hence, IntCal13 was used in Chronomodel. The ages appear in stratigraphic order and exclude outliers (see section 3.1 below for details).**

All calibrated ages shown in Fig. 2 belong to the Çatalhöyük mound period, except the two bottom-most samples that pre-date the mound. So, if we consider all ages from the mound, two sets of samples can be distinguished: one sample set at the bottom of the mound (from sample OxA-9893 to sample 109993) dates from between ~7350 BC and ~7050 BC, and a second set (from sample OxA-9776 to sample 103134) that dates from between ~7200 BC and ~6700 BC. To put things simply, the ages are rather imprecise because they correspond to two plateaus of the calibration curve; and, according to the unmodelled data, the mound was built after 7350 BC and stopped being occupied before 6700 BC. However, Bayliss et al. (2015) concluded from their chronological model that deposit from the mound started to accumulate between 7165 and 7085 BC (at the 95%



confidence level), which is in striking contrast with the measurements from the mound itself. This modelling result then has significant archaeological implications regarding the appearance of pottery and sheep domestication in the area. Furthermore, it raises questions about a 4-century time gap between the onset of Çatalhöyük and nearby archaeological remains found in Boncuklu (see Bayliss et al., 2015, and references therein). The questions we want to address is (i) that of the consistency of the modelling outcome, (ii) to determine whether other models would provide a different inference, and most importantly (iii)

discuss the reasons explaining the potential discrepancies obtained with various models.

### 3. Results

### 3.1. Loess plateau dated by OSL

Fig. 3 shows the output of BayLum, i.e., the modelled sequence obtained after 10 million iterations (note: convergence of the MCMC chains was assessed using the Gelman-Rubin statistic; convergence only required 5,000 iterations when the ordering

constraint is not imposed. The fact that so many iterations were required when modelling ages under stratigraphic constraints already suggests that the results should be taken with caution).

For clarity, on this figure the average age of the 10 samples estimated with BayLum, without imposing the stratigraphic constraints, is indicated (14.7 ka, instead of 14.6 ka estimated with the ages published by Stevens et al., 2018; we regard this 1% difference as insignificant). Visually, as expected the ages now appear in stratigraphic order; in addition, the ten ages seem

to form a smooth sequence of continuous sediment accumulation, in sharp contrast with Fig. 1. One may also note that only 5 ages agree with the average of unmodelled ages (NB: by unmodelled, here we refer to ages estimated without imposing the stratigraphic constraints) at the 68% credibility level; 4 at the 95% level, and 1 age does not include 14.7 ka at the 95% level. In other words, it seems that imposing stratigraphic constraints leads to ages in poor agreement with the measured data.





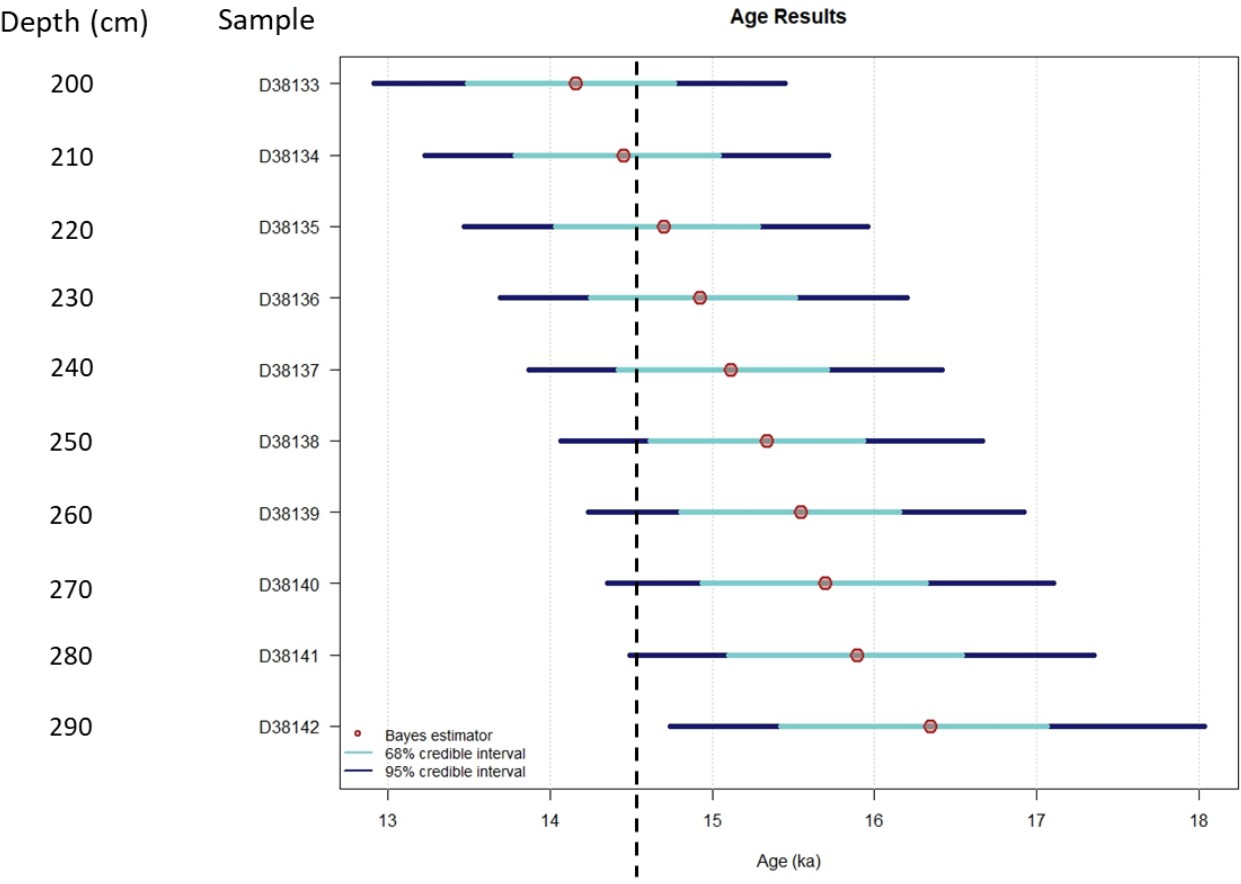

**Figure 3. Jingbian OSL ages estimated with BayLum, taking into account stratigraphic constraints and systematic errors.**


Figure 4 shows four series of OSL ages estimated with BayLum using stratigraphic constraints and systematic errors. To investigate the influence of sampling density, we made the number of samples vary from 2 (keeping only the top most and bottom most samples) up to 10 (case shown in Fig. 3). In each case, we kept the top most and bottom most samples. Intermediate calculations included 4 and 7 samples, respectively. The convergence of MCMC chains was assessed using the

Gelman and Rubin statistic; it is worth emphasizing here that calculation times (i.e., the number of iterations) needed to reach convergence increased exponentially with the number of samples – well beyond the additional time required to account for the greater number of variables. We attribute this feature to the decreasing agreement between the input data and the modelling output: BayLum struggles to find an appropriate solution combining the data and the stratigraphic order, which is not surprising when considering the measured ages (Fig. 1). When looking at Fig. 4, it appears that the sampling resolution strongly affects





the modelling results; more precisely, the spread effect is increased when the number of samples is increased, as was expected based on the study of Steier and Rom (2000). Indeed, the difference between the two extreme ages (top and bottom) appears to increase when intermediate samples are analysed. Moreover, when the number of samples included in the modelling is increased, it appears that the credible intervals become more and more shifted from the age intervals estimated independently.

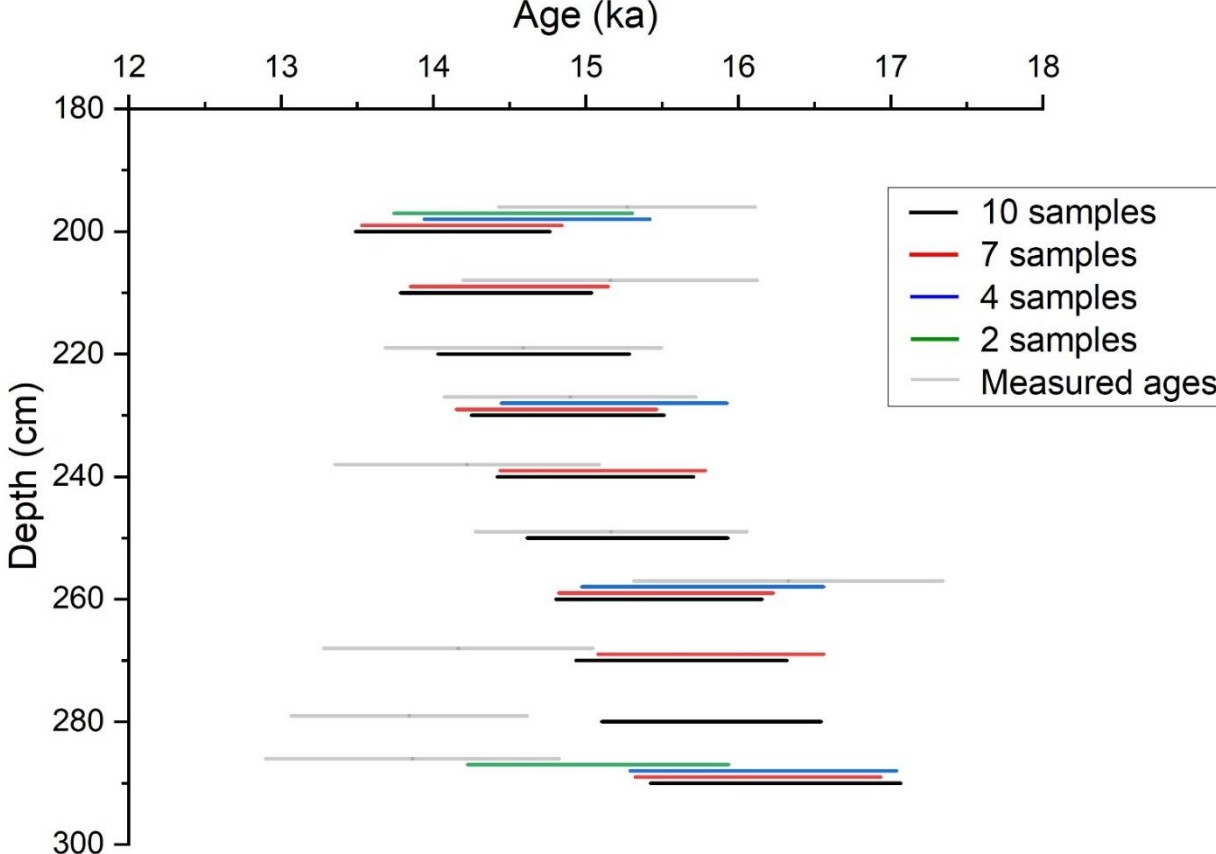

**Figure 4. Series of OSL ages estimated with BayLum, taking into account stratigraphic constraints and systematic errors. In all series, the top most and bottom most samples are included in the calculation. However, the total number of samples is varied from one calculation to another, from 2 up to 10. Note that for clarity, the y-coordinates of each sample are slightly shifted when the total number of samples included in the model is varied. The true depth values correspond to the black lines.**





To investigate this issue further, we looked at the joint probability, *i.e.*, the probability that the ages modelled in stratigraphy fall in the 95% C.I. of the ages estimated independently. To do this, we use the MCMC sample used to estimate the posterior probability density of the ages. We then count the fraction of iterations for which all individual ages belong to their initial 95% C.I., estimated without imposing stratigraphic constraints. Ideally, an increase in the number of samples would lead to a gain in the precision of the statistical inference, i.e., the length of the C.I. should decrease whilst the model output stays consistent with the data – so the joint probability should stay close to 1. In other words, the joint probability ranges from 0 to 1 and is a measure of the agreement between the model output and the data used as input – 1 indicating perfect agreement and 0 very poor agreement (Nb: this joint probability is similar to OxCal's agreement index; Ramsey, 1995). Fig. 5 shows the evolution of the joint probability as a function of the number of samples included in the calculation, following the same procedure as in Fig. 4; in Fig. 5, we normalised this joint probability by $0.95^n$, where n is the number of modelled samples (therefore, if the stratigraphic model yielded identical results compared to the unmodelled data, this normalised probability would be equal to 1). The joint probability of ages decreases when the number of samples modelled in stratigraphy is increased, demonstrating that the model output becomes less and less consistent with the input data. Of course, such a situation might occur if data is inconsistent with the assumption underlying the model, for example if outliers are present, or if – for some reason – ages decrease with depth. However, in our case the data simply suggest a fast accumulation of sediment, all samples being deposited around 14.7 ka. As pointed out by Steier and Rom (2000) in the case of radiocarbon, it appears that modelling of OSL ages in stratigraphy using BayLum, and thus Eq. (1), leads to the same spread effect.



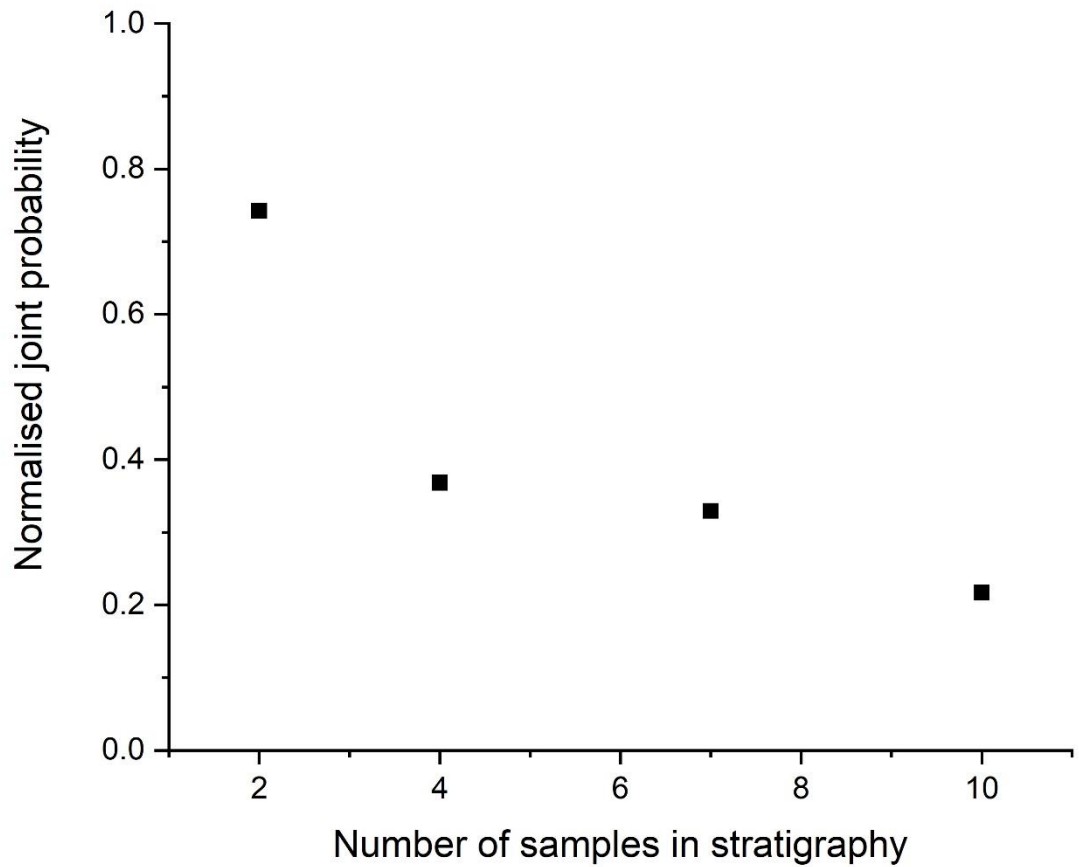


**Figure 5. Normalised joint probability of the BayLum model output and the initial age estimates. The joint probability corresponds to the probability that each iteration of the MCMC, when calculating ages in stratigraphic constraints, falls in the 95% C.I. of the ages estimated independently, i.e., without imposing stratigraphic constraints. For the model performance assessment, we normalise this probability by $0.95^n$, where n is the number of samples (therefore, if the**
**stratigraphic model yielded identical results compared to the unmodelled data, this normalised probability would be 1).**

Finally, we also estimated the duration between deposition of the first and last samples considered here, as a function of the number of samples included in the model (in a similar fashion as for Figs. 4 and 5). To this end, we used the function
PhaseStatistics of the R package ArchaeoPhases (Philippe and Vibet, 2020). Based on visualisation of the data (Fig. 1), we expect a short duration (compared to measured uncertainties). However, it appears on Fig. 6 that an instantaneous deposition





event (i.e., a phase of null duration) is only included in the 95% C.I. of the duration in the case where only 2 samples are included in our BayLum model. Overall, both the range of values and the length of the C.I. increase with the number of modelled samples. In other words, when including more samples, and so more information, the precision of the estimate

decreases and the short duration values become less and less credible, which is in striking contradiction with the data.

To summarise, the model of stratigraphic constraints (Eq. 1) implemented in BayLum is not robust; neither is it precise (since the precision of the estimated duration increases with the number of modelled samples), nor accurate (since short phases are not amongst the most credible values, contrary to what the data suggest).

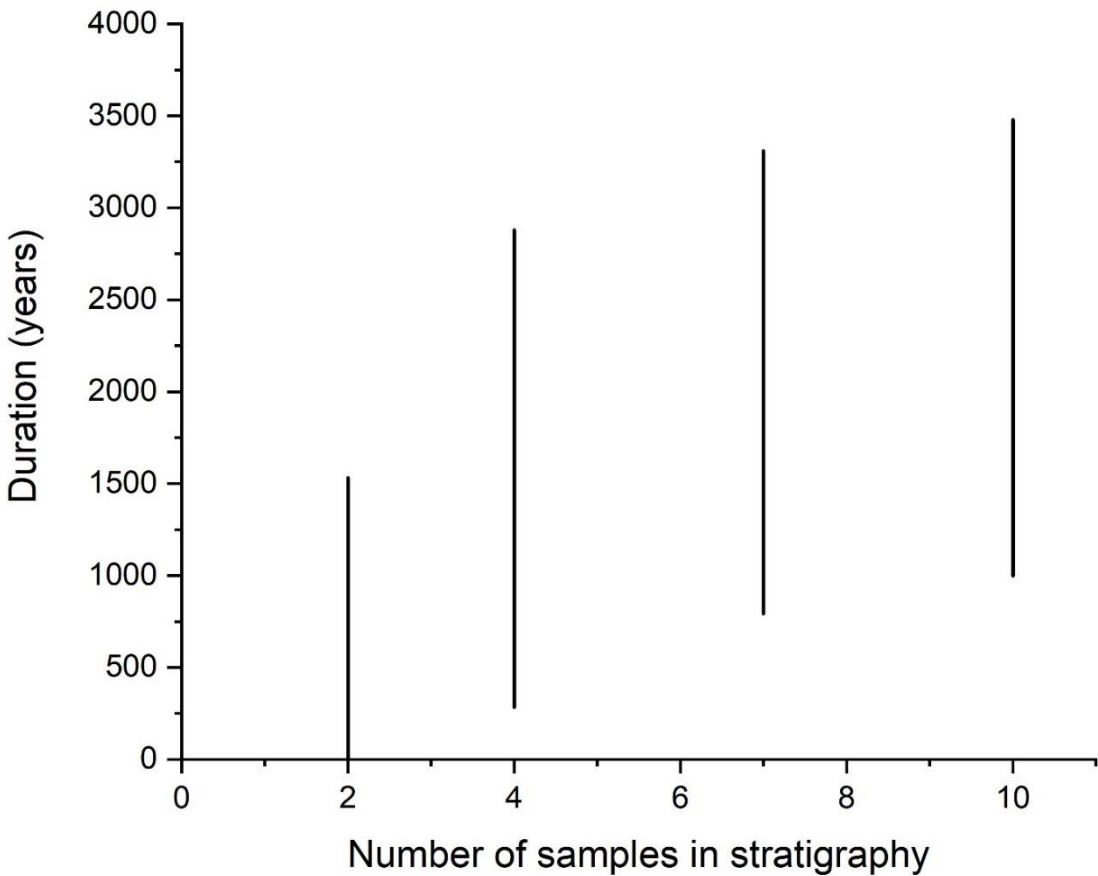

**Figure 6. 95% credible interval of the duration between the first and last sample deposition, based on the ages shown in Fig. 4. It appears that the range of credible values increases when the number of samples modelled in stratigraphy is increased. In addition, the length of the 95% credible interval is also increased, which corresponds to a counterintuitive loss in precision when more data are used for the estimation.**



### 3.2. Radiocarbon dating at Çatalhöyük

#### 3.2.1. OxCal

Fig. 7 presents the sequence of 46 radiocarbon ages modelled with OxCal. Seven ages appear as outliers and are shown in red on this figure, as in the original study. All seven outliers correspond to hackberry seed samples (*Celtis sp.*), except one bone from a stillborn sheep (UCIAMS-109995). The reason why these samples yielded anomalous ages was unknown in the original study and it still is today. In the following, unless explicitly stated otherwise, these samples are excluded from our dataset.





**Figure 7. OxCal model for the sequence of Çatalhöyük. Empty curves correspond to calibrated radiocarbon ages, while filled curves correspond to modelled ages according to the stratigraphy. The phase structure reproduces that of the initial publication (Bayliss et al., 2015). Red and brown: outliers; grey: phase boundaries; green: all individual ages consistent with the model. Note that the boundary 'Start Mound' occurred after the two bottom-most dates, which are *terminus post quem*.**




The concentration effect is visible at the bottom of the sequence, where large parts of the probability density distributions of the calibrated radiocarbon ages are excluded of the model inference (between samples OxA-9893 and OxA-23251). Yet, only two ages (the two bottom most samples, that predate the onset of the mound) constrain these ages; and the two bottom-most are affected by very large uncertainties due to a calibration plateau. These two bottom-most samples are PL-980252A, whose age lies outside the calibrated age of all samples above – so it does not bring any really useful information regarding the start of the mound; and sample AA-27982, whose calibrated age (defined here as its highest posterior density (HPD) interval) partly overlaps the density distributions of samples above. The model completely removes the possibility that the age of sample AA-27982 is greater than 7300 BC, although approximately one third of the HPD interval extends beyond this date. This concentration effect is even more visible on Fig. 2 of Bayliss et al. (2015)[1]. We see no reason to this behaviour other than a modelling artefact. Simply looking at the calibrated data, one would think that the start of the mound is plausible at 7300 BC, since 8 samples from the mound itself include this date in their 95% credible interval (C.I.).

The spread effect is also quite clearly visible, on the top part of the sequence (from sample OxA-9776 upwards): essentially, all these samples have indistinguishable ages, because of the plateau of the calibration curve. To put it in simple words, most – if not all – of these samples have a calibrated credible age interval spanning between 7100 BC and 6700 BC. Saying that some are older than others cannot reasonably end up in saying that some of these samples are very precisely dated to between 7100 and 7000 BC, which is the outcome of the OxCal model. In our view, it does not appear sensible to hope that a stratigraphic model may resolve calibration issues, and in particular plateaus in the calibration curve.

### 3.2.2. Chronomodel

Fig. 8 presents the chronological inference obtained with Chronomodel for the bottom part of the sequence (note: replicate measurements of a sample were included in one event). It should be noted here that including more samples on this figure would make the curves and text unreadable. Therefore, we chose to represent the bottom (rather than the top) part of the sequence, because this is the part which has the most important archaeological implications (at least as of Bayliss et al., 2015). It should be emphasized that most, if not all of our following observations and comments are illustrated by this figure, even though it only partially reflects our modelling results.

As expected, Chronomodel easily accommodates outliers thanks to the event model. However, the spread effect is rather pronounced: quite quickly along the stratigraphy, the age probability densities of subsequent samples are indeed not overlapping, contrary to what is observed on Fig. 1 showing the calibrated ages. This is not surprising because the event model, which is the modelling basis in Chronomodel, inflates analytical uncertainties and so is expected to increase the modelling artefact (see introduction).

---

[1] Marginal differences seem to occur between the OxCal model of Bayliss et al. (2015) and ours, presumably because we used IntCal20 (Reimer et al., 2020) instead of IntCal13 (Reimer et al., 2013) used by Bayliss et al. (2015) to calibrate radiocarbon ages.







**Figure 8. Part of the chronological model obtained with Chronomodel for the sequence of Çatalhöyük, showing the 17 bottom-most samples.**

### 3.2.3. BayLum

Before using BayLum, all radiocarbon ages obtained from hackberry endocarps were excluded from our analyses, as well as sample UCIAMS-109995, since they all appeared as outliers in the OxCal model. It should indeed be noted that the algorithm for outlier detection in BayLum appears to be less efficient than that of OxCal. However, the chronological inference obtained with OxCal also ignores these outlier ages, so the comparison between chronological inferences obtained with OxCal and BayLum remains valid.



Fig. 9 shows the output of BayLum for the 40 remaining samples, obtained after 500,000 iterations. Here also, the 'spread effect' is visible – to a comparable extent to that observed with OxCal, and so less than with Chronomodel.

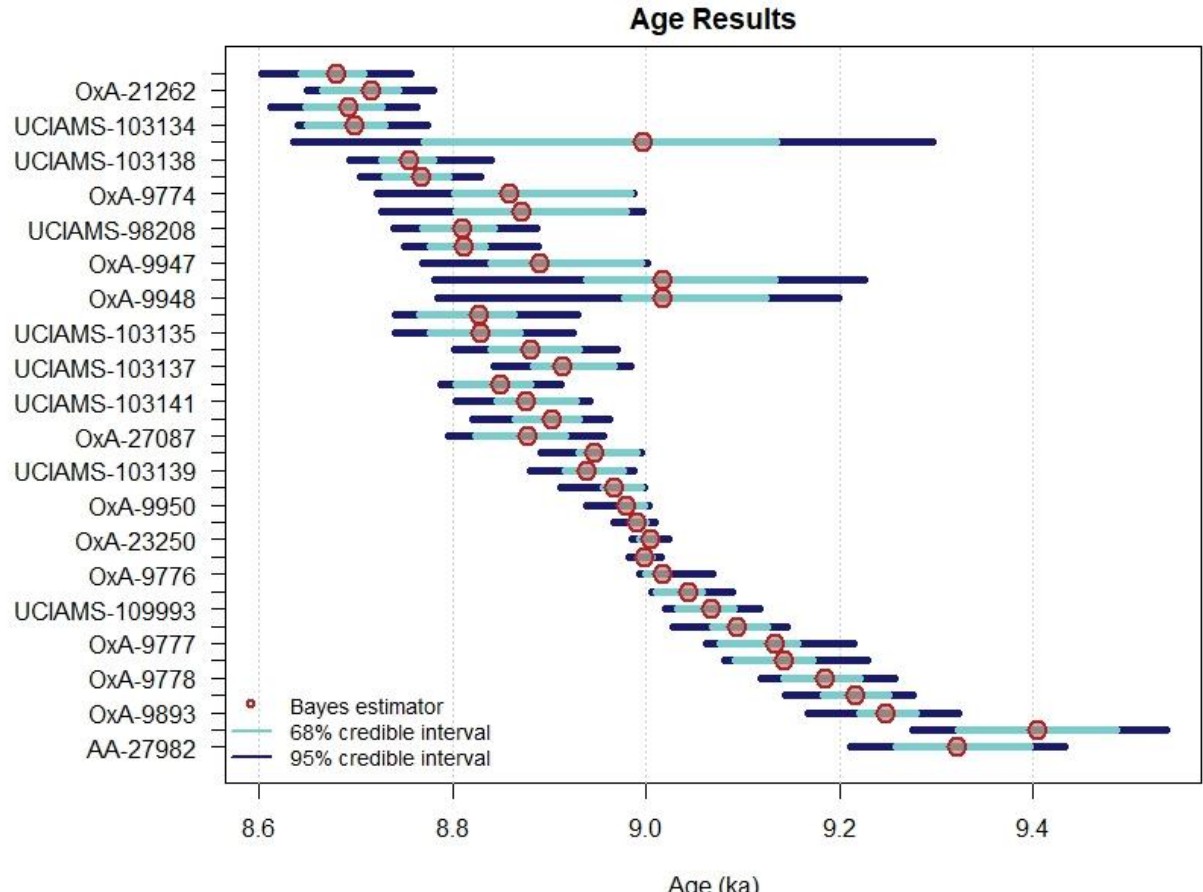

**Figure 9. Ages for the sequence of Çatalhöyük modelled with BayLum. Note that the output of BayLum only indicates the sample labels for one every two samples (the reader is referred to Table S1 and Figs. 1, 2, 3 for the identity of the other samples).**



## 4. Discussion: Stratigraphic constraints across different models and methods.

Steier and Rom (2000) identified issues posed by Eq. (1) when used to model radiocarbon ages using OxCal. With the first sample set of the present study, we could illustrate similar problems arising when using the same equation, but using BayLum to model OSL samples in stratigraphy.

### 4.1. Çatalhöyük ages

The rest of the first part of this discussion is, therefore, dedicated to the comparison of models when dealing with the same
dataset – the radiocarbon-dated sequence of Çatalhöyük. Fig. 10 presents the comparison between OxCal, Chronomodel and BayLum. Not knowing the true age values since no historical sources attest to the occupations of the site of Çatalhöyük, it is difficult to make definitive statements. Nevertheless, several remarks can be made.

**Figure 10. Comparison of radiocarbon ages modelled with OxCal, Chronomodel and Baylum (95% C.I.).**




A first observation is that, for four samples towards the bottom (from OxA-9893 to OxA-9777), there is no overlap between the 95% C.I. obtained with OxCal and neither that obtained with BayLum nor that estimated with Chronomodel. In other words, it is very likely that at least one model provides flawed results, i.e., the true age value lies outside the age interval

estimated by at least one model.

Second, some samples display extremely small 95% C.I. age intervals, for example UCIAMS-109991 and UCIAMS-109992 have an age uncertainty corresponding 23 and 24 years, respectively, according to OxCal – which corresponds to ~0.05 % uncertainty (expressed as one quarter of the 95% C.I. divided by the age, in our case ~9,000 years). Note that the length of these 95% C.I. is only slightly larger according to BayLum: 56 and 33 years, respectively. In a period characterised by a plateau

of the calibration curve, this observation should raise a flag about the validity of the model – such a precision level seems doubtful.

A third observation is that ages estimated with Chronomodel display significantly larger credible age intervals, i.e., greater uncertainties, compared to BayLum and OxCal. In other words, Chronomodel appears to lie on the cautious side of things – which is perhaps an advantage given our first observation, namely that at least one model gets (at least part of) the chronology

wrong.

### 4.2. Çatalhöyük: start of the mound (onset of occupations)

Bayliss et al. (2015) were particularly interested in the chronology of the start of the mound, which has important archaeological implications. From this perspective, it is interesting to compare the radiocarbon calibrated ages with the statistical inferences obtained with OxCal, Chronomodel and BayLum (Figs 11 and 12).

In OxCal, the boundary 'start mound' is dated to between 7144 and 7079 BC (NB: we base our statements on the 95% C.I. in this paragraph). According to Chronomodel, the earliest sample (sample OxA-9893) from the mound, i.e., the sample that is used to determine when occupations started, is dated to between 7487 and 7221 BC (95 % C.I.). Finally, according to BayLum, the age of this sample is dated to between 7373 and 7217 BC. Let us now compare these statistical inferences with the calibrated ages obtained for samples from the bottom of the mound. The bottom-most sample from the mound is sample OxA-9893,

dated to 7326-7049 BC (calibrated age). As a consequence, the start of the mound cannot, at the 95% credibility level, be earlier than 7326 BC (NB: in principle it can, because one cannot be sure that this sample is the very first remain left by people occupying the site; but any modelling solution earlier than that would by definition extrapolate the data). It is important to reiterate here our previous observation that this age interval is very similar to the age intervals of the seven overlying samples (Table 1). Based on this sample set, one may assert that the start of the mound cannot – at the 95% credibility level – have

occurred after 7077 BC (this is the younger age limit for sample UCIAMS-109994).



**Table 1. Calibrated ages of the eight earliest samples from the mound of Çatalhöyük, the two samples predating the mound (in italic), and comparison with statistical inferences concerning the start of the mound according to OxCal, Chronomodel and BayLum.**

| Sample or start of the mound | 95% C.I. for the calibrated age or modelled start of the mound (in years BC) |
|---|---|
| OxA-23251 | 7314-7048 |
| UCIAMS-109993 | 7316-7061 |
| OxA-9892 | 7326-7047 |
| OxA-9777 | 7326-7052 |
| UCIAMS-109994 | 7335-7077 |
| OxA-9778 | 7461-7076 |
| OxA-23252 | 7328-7075 |
| OxA-9893 | 7326-7049 |
| *AA-27982* | *7471-7047* |
| *PL-980525A* | *7589-7186* |
| **Start of the mound (OxCal)** | **7144-7079** |
| **Start of the mound (Chronomodel)** | **7487-7221** |
| **Start of the mound (Baylum)** | **7373-7217** |





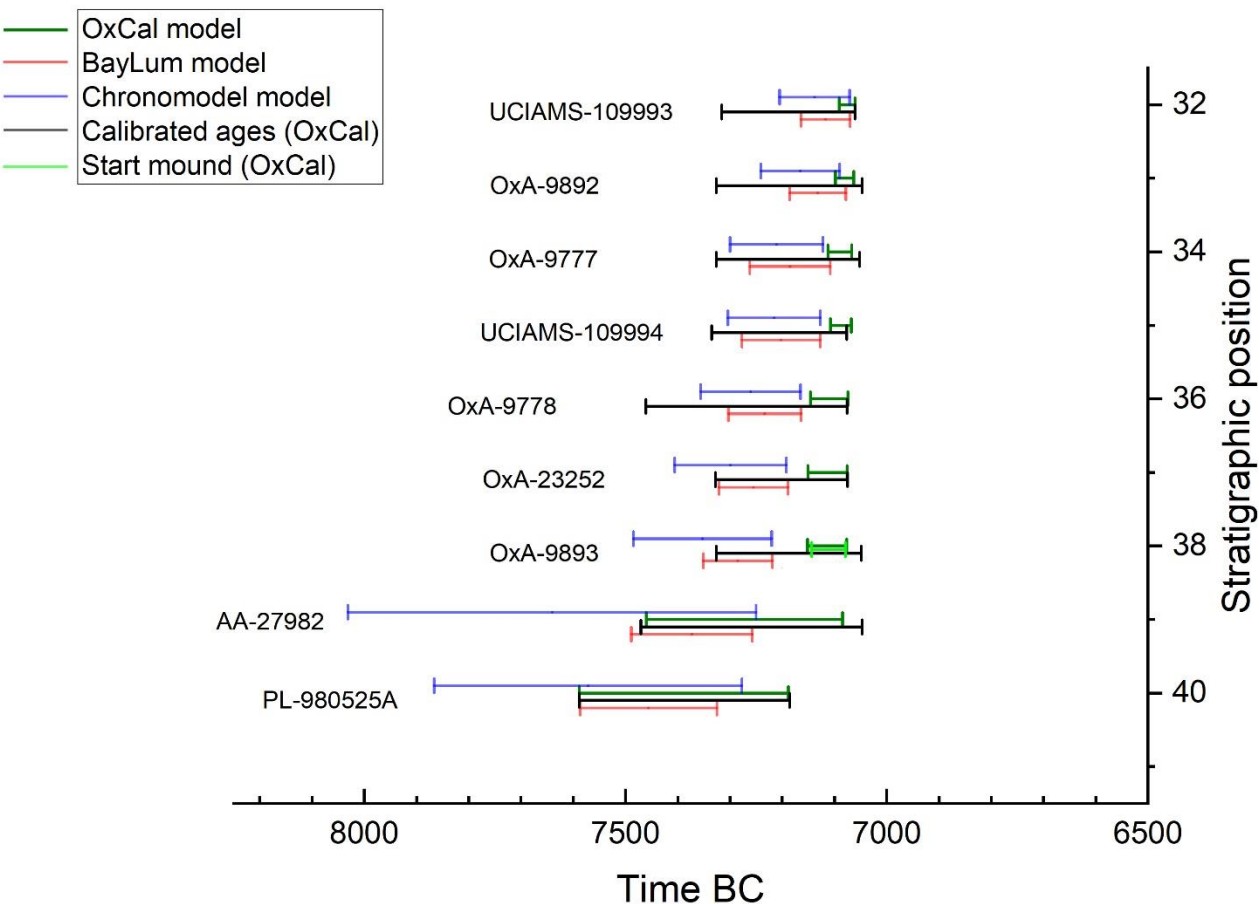

**Figure 11. Bottom of the sequence: calibrated ages (using OxCal) in comparison with ages modelled with OxCal, Chronomodel and BayLum. Sample OxA-9893 (third from the bottom) is the oldest from the mound and thus serves as a proxy for the start of the mound in BayLum and Chronomodel. The onset of the mound estimated with OxCal is**
**also shown.**

From this perspective, at the 95% credibility level, (i) Chronomodel appears to suggest an age interval that is too old – about 50% of this interval correspond to ages older than 7326 BC; (ii) The date for the start of the mound inferred with BayLum is the closest to the age interval, although both BayLum and Chronomodel exclude a 'recent start' (after ~7220 BC); (iii) OxCal
completely excludes an 'old start' for the mound, in striking contradiction with the data (see Fig. 12). Indeed, all individual ages from the bottom most layer have part of their 95% C.I. extending beyond 7144 BC. The sample that appears most recent (OxA-23251), in the sense that the older boundary of its 95% C.I. is the most recent date of all C.I. from this layer, has a 95%




C.I. extending from 7314 to 7048 BC. Yet, according to OxCal, at the 95% credibility level, the phase cannot have started earlier than 7144 (the 95% C.I. is 7144-7079 BC). In our view, this is clear evidence that, in this case, OxCal provides results

that poorly reflect the available chronological information, i.e., the calibrated ages. It is interesting to look here at the two samples from below the mound; one might indeed assume that such samples would explain why OxCal excludes early ages for the start of the mound. However, their ages (7471-7047 BC and 7589-7186 BC) are not precise enough to exclude an early start of the mound. In other words, one is typically here in the case of an artificial gain in precision obtained with OxCal (as already highlighted by Steier and Rom, 2000), in contradiction with the data.

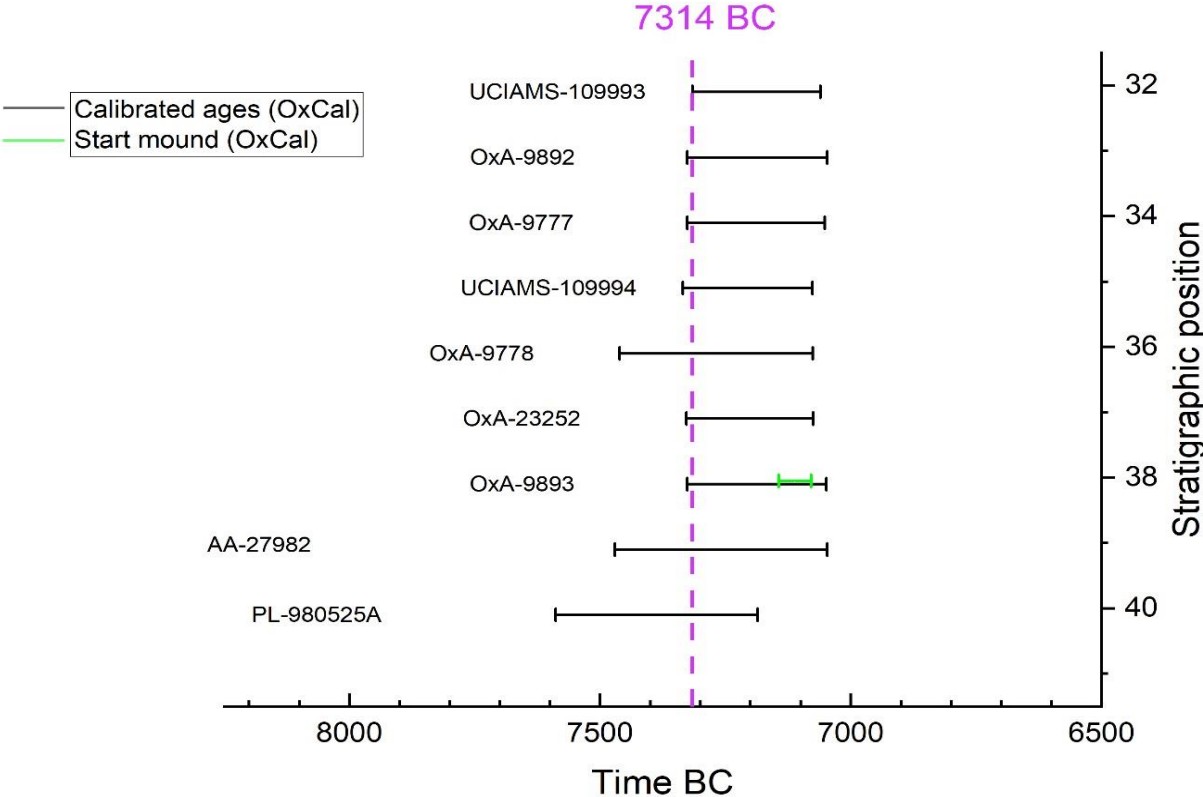


**Figure 12. Comparison of calibrated radiocarbon ages (95% credible intervals in black) with the start mound boundary, as estimated with OxCal (light green). All samples on this figure (except AA-27982 and PL-980525A, at the bottom) belong to the lowermost layer of the mound, i.e., correspond to the layer yielding the earliest human occupations of the site. Samples AA-27982 and PL-980525A pre-date these human occupations. The dashed purple line**

**corresponds to 7314 BC. This is the earliest date all individual ages from the lowermost archaeological layer are consistent with, at the 95% credibility level. Thus, excluding a start around or shortly after this date seems very unreasonable.**





### 4.3. Comparison of models: overview

Table 2 summarises our observations in the form of a list of modelling artefacts: all three models (OxCal, Chronomodel and BayLum) are affected by the spread effect due to what appears to be poor modelling of stratigraphic constraints (Eq. 1); OxCal is affected by the concentration effect because of the uniform phase model (Eq. 2); Chronomodel adds extra uncertainty that is not quantified during measurement (note that this may actually be an asset if analytical uncertainties were underestimated, or if an unknown source of error is at play).


**Table 2. List of modelling artefacts that may be encountered in chronological modelling, based on our study of radiocarbon dating at Çatalhöyük.**

| Modelling artefact | Source of the problem | Affected software | Consequences |
|---|---|---|---|
| Spread effect | Ordering (or stratigraphic) constraints (Eq. 1) | OxCal Chronomodel BayLum | Estimation bias<br>Artificial reduction of uncertainties<br>Probability densities of undistinguishable ages become disjointed |
| Concentration effect | Phase model (Eq. 2) | OxCal | The tail of probability densities of extreme samples is cropped, all samples inside a phase become artificially clustered |
| Extra uncertainty | Event model (Lanos and Philippe, 2017) | Chronomodel | Age uncertainties are artificially increased |

By definition, all models are wrong – a model is a mean to simplify experiments (real life) so that a theory can be designed. In addition, modelling choices come into play. As a result, no particular model is fundamentally bad or good, thus the purpose of this paper is not to tell readers to stop using all the above-mentioned models; it rather is to illustrate some modelling artefacts that should be given attention. In our second case study, let us first examine the data available to define the start of the mound (the intervals in black in Fig. 6 represent the 95% credible interval of all ages relevant for this question). We observe (i) that,

at the 95% credibility level, the ages preceding the start of the mound fall between 7589 and 7186 (sample PL-980525A), and between 7471 and 7047 BC (sample AA-27982), with a near-uniform probability density (so all ages within this interval have similar probabilities; see Fig. 2); (ii) all 95% C.I. for the samples from the first layer of the mound are very similar (Fig. 6)





and are included in the interval [7461, or 7335 if one excludes ample PxA-9778; 7047] BC. From these observations, excluding a start of the mound around 7314 BC seems very unreasonable, because all samples from the archaeological layer are consistent with this age (Fig. 12), and because the only samples that might exclude such an early start, i.e. the samples below the archaeological layer, are perfectly consistent with an earlier age (with a probability of ~35% for the youngest sample AA-27982).

It follows that none of the three tested models gives a solution to the question of the start of the mound that is consistent with the data: OxCal because it gives a way too recent (and too precise) date, Chronomodel because it gives a high probability to a start of the mound much earlier than any measured age suggests – while excluding a recent start of the mound (compared to measured age ranges), and BayLum for the same reasons as Chronomodel (although to a marginal or negligible level regarding the extrapolation towards older ages). As frustrating as it may be, in our view none of the tested models can tell us anything better than the actual data themselves: the mound started to build between 7326 and 7077 BC. Therefore, the time gap identified by Bayliss et al. (2015) between the occupations in Boncuklu (dated to no later than 7500 BC – but even there, this date is open to discussion since the site's latest layers were truncated; see discussion in Bayliss et al., 2015) and Çatalhöyük may not have been as long as the 400 years claimed by these authors. At least, the data do not support their claim of a late (~7100 BC) start for the occupations of Çatalhöyük.

## 5. Conclusion

All chronological models tested in this study – namely OxCal, Chronomodel and BayLum – are affected by the spread effect. This effect tends to create an estimation bias when modelling a short period with large uncertainties, i.e., when age uncertainties are large compared to the actual time differences between samples. This effect is particularly visible for the sequence of Çatalhöyük. Consequently, all three models provide statistical inferences that are, to some degree, inconsistent with the measurements. Of these three models, it appears that for Çatalhöyük, it is Baylum that provides the results that are more consistent with measurements. Notwithstanding the generality of this statement, our results suggest that other features of OxCal and Chronomodel explain their poorer performance: the phase model in OxCal, which concentrates ages inside a phase; and the event model in Chronomodel, which leads to increased uncertainty and thus better accuracy/robustness, but which increases the spread effect. This being said, BayLum clearly leads to aberrant results when modelling sequences of indistinguishable ages, as exemplified with the OSL dataset from Jingbian.

Overall, this article should be viewed as a warning message: when testing any chronological model, it is of utmost importance to compare the model outcome with the input data. Statistical artefacts that should be given special attention are the spread effect (common to OxCal, Chronomodel and BayLum), the concentration effect when using the phase model implemented in OxCal, and unwanted loss of precision when using the event model implemented in Chronomodel. Obviously, chronological modelling aims at improving the statistical inferences that can be attained from a given dataset – but it should not lead to inconsistent results.



455 Finally, comes the following fundamental question: what are we using Bayesian chronological models for? The general consensus it that they are implemented to make use of prior observations, such as stratigraphy, to refine the precision, accuracy and robustness of numerical chronologies. For example, one may hope to gain in precision without losing accuracy when including ordering constraints, by somehow correcting measurement errors. Our study shows that this goal is difficult to reach and that using models to correct measurements appears to be dangerous. Reducing uncertainties may indeed lead to loss of

460 robustness and to conclusions that are inconsistent with the data, at least for the examples studied here.

**Acknowledgements**

This study received financial support from the European Research Council, through ERC starting grant Quina World (#751893) awarded to G. Guérin.



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
