# Peer review of "The conflict between sampling resolution and stratigraphic constraints from a Bayesian perspective: OSL and radiocarbon case studies"

_EGUsphere, 2025_

## Referee Comment (RC2)

Review of

**The conflict between sampling resolution and stratigraphic constraints from a Bayesian perspective: OSL and radiocarbon case studies**

28th June 2025

**Overview**

This paper presents two worked case study examples that show three different modelling approaches can lead to strikingly different results when applied to the same data.

I think this is an interesting result, and important for readers who might be using these models to understand, but I do feel that the current exposition is really quite unclear – in terms of sufficiently describing what the different models do; their interpretation; and the potential reasons for any differences in the resulting inference.

They argue that the differences they see between models are due to the way that they handle stratigraphic ordering, but it is not entirely clear to me that it is solely this – as they also seem to argue that the models implement this stratigraphic information in very similar ways.

If that is the case, then I would expect similar results across the models. Instead it suggests to me that either they actually handle stratigraphy quite differently, or that there are more fundamental differences between the three models (which are not explained) that effectively lead to quite different modelling assumptions; or that (some of) the models have perhaps not converged correctly.

Specifically, they compare:

1. BayLum when modelling OSL dates
2. OxCal, BayLum and ChronoModel when used to analyse a selection of 14C dates from Catalhoyuk

As I said, I think the overall manuscript has highly useful and valuable content for the community and I would recommend publication, but IMO the overall narrative and level of clear explanation really does need to be improved if it is to be of substantial use to the relevant community.

**Major Comments**

**Explanation of the Models**

There needs to be much more, and understandable, detail on the mechanics of the three actual models. Currently, there is quite a lot of written text (which I feel is hard to parse and repetitive in places) but I was not particularly clear on how any of the three approaches actually modelled the data – in particular BayLum or ChronoModel as these are less well-known to me.

I do not think that in this instance, where you are arguing the models are inconsistent with one another, it is sufficient just to refer to the original papers. One needs to understand why the models might be presenting different results to one another (and that must either be because they have different underlying statistical modelling approaches, or that the MCMC they all seem to use hasn't

converged properly (which is a serious issue with using MCMC on an ordered parameter space as it becomes highly multimodal).

I understand you cannot describe the entire approach for each, but what are the actual mathematical/statistical models for each that are fitted (what is the likelihood, what is the prior, …). In my view this should be done through short mathematical equations with a suitable notation, not long passages of text which aren't sufficiently explicit.

If it is as simple as just some phase models and stratigraphy I would hope/assume that this can be provided in a couple of general equation (see e.g., Nicholls and Jones, 2001, for a complete and general explanation of the stratigraphic phase model as I presume is fitted in OxCal to Catalhoyuk).

**Comparison with Nicholls and Jones (2001)**

It is not clear to me that the authors have entirely understood this paper as they seem to argue against the point I see it as trying to make several times (e.g., line 51) while then citing it as an example of an approach to address the problem of *under* spreading as they see it over. Importantly, I do not see this paper as making a particular statement about using a uniform phase within a boundary. It is about a different aspect – the prior on phase boundaries.

My reading/understanding of Nicholls and Jones (2001) is that their main thrust is to argue that if you do not use a sensible prior then you get erroneous **overestimation** of the spread of stratigraphic calendar ages (i.e., dates that are too far spread). Hence they are arguing precisely that you should compress/shrink the calendar ages. This paper evidently needs to be referenced but I think in a very different way.

Nicholls and Jones' specific, recommended, prior actively aims to penalise the spread of phase boundaries. This problem of over-spreading is well known and theoretically evidenced (e.g., by Stein, 1956). Nicholls and Jones demonstrate this in the context of 14C calibration convincingly using Bayes factors. They also provide a discussion of what underlying depositional model leads to their chosen prior which penalises the spread, and an argument it is more sensible than the model which generates a constant prior density. They are therefore arguing against the idea of this paper (i.e., they are making a case things should be compressed!).

Nicholls and Jones (2001) consider a model that consists of ordered phase layers (with adjoining boundaries $\psi_0, \ldots, \psi_M$) and within each layer they assume there are a set of samples with calendar ages. They assume no ordering within the samples inside a layer. This seems to be precisely the model you fit to Catalhoyuk.

Specifically, they assume that the $N_i$ $^{14}$C samples within a specific layer/phase $i$ (i.e., the samples between calendar ages $\psi_{i-1}$ and $\psi_i$) have calendar age $\theta_{i,1}, \ldots, \theta_{i,N_i}$ and are uniformly distributed in calendar time through the layer (i.e., there is no information, or prior, placed, on their relative ordering). The ordering is really only placed on the phase boundaries, i.e., $\psi_i > \psi_{i-1}$

Throughout their paper, they propose a completely consistent prior on the individual sample calendar ages $\theta_{i,j}$ (conditional on the phase boundaries). This is a uniform prior because they don't assume any ordering of the samples within the layer.

Their argument is entirely concerned with what prior you should place on the phase boundaries. They propose two – a constant prior that does not penalise the overall range/span of all the layers which they argue is actually highly informative on the overall span and leads to overspread inference; and

one that does, i.e., operate to reduce the overall span $\psi_M - \psi_0$ which they argue (with theoretical support) works better.

Which it the bit you are arguing against? The uniform phase component of the model within each layer? Or the prior on the spacings of the phase boundaries. These are quite different things.

As far as I am aware, Nicholls and Jones (2001) do not discuss, or argue for/against the uniform prior on the samples within each layer – this is taken as an assumption of both their priors/models.

If I am correct then I think this therefore means that some needs some significant rewordings are needed in the paper. There is stuff about uniform phase models being needed to reduce erroneous over-spread (where the samples do arise from such a model) but it I'm not sure this is Jones and Nicholls (2001).

**Assessment of Model Appropriateness**

I entirely agree with the authors that all users should not treat statistical models as black-boxes, but rather ensure/investigate that the assumptions made within those models are valid for their example/analysis. Also that there is no such thing as a non-informative prior (and that the choice of prior can have large consequences on inference). This is something which is far too frequently overlooked, especially in Bayesian analyses (which will always give you an answer even if it doesn't make sense).

However, I am a little concerned by the idea/conclusion which I feel goes too far in suggesting that you should perhaps ditch modelling altogether, and that you can conclude the models are wrong primarily because they don't overlap with selected independently-calibrated 14C dates. There is a large literature (backed up by theory) which tells you that independent estimation of any random variables is not a statistically-valid approach and leads to overly-spread estimation (going back to Stein, 1956). We should not encourage people to go back to that.

Of course, one should be concerned by the differences between the inference provided by these models but I think the reasons are more nuanced than you present. As you say none of the models are likely correct – but then neither is independent calibration of each sample. Presumably some of the models fundamentally have components which you subjectively might not support in their construction/prior. I feel you need to draw that out (which would be easier if the actual statistical modelling equations were explicitly laid out)

**Technical/Minor Comments**

Equation 2 seems wrong to me. Should the numerator on the RHS be 1; and the product moved outside of the fraction? Currently the RHS numerator repeats the LHS – which is circular.

BayLum uses IntCal13 – I would suggest that you need to show/state that this does not lead to the differences in inference. It shouldn't as IntCal13 and IntCal20 are very similar in the period of Catalhoyuk but you need to explain/show this (e.g., by providing  plot of the two IntCal curves alongside one another in this interval).

I commend the authors on making their code available but it would be nice if it was on Zenodo/Github rather than simply text in an Appendix.

---

## Author Comment (AC1)

Dear Editor(s) and Authors,

I have read the manuscript 'The conflict between sampling resolution and stratigraphic constraints from a Bayesian perspective: OSL and radiocarbon case studies' in detail. It is a well structured and well-written manuscript, and the scope fits Geochronology. You address a standing topic and issue in age-depth modeling, and your contribution is relevant. While your conclusions are not entirely new, these are based on real and well selected datasets, and your manuscript is a welcome part for the scientific literature. **I clearly support publication in Geochronology after revisions.**

We would like to thank the referee for such positive feedback. We would also like to emphasize that the review comments provided below are – in our view – very constructive, and we thank the referee for such a thorough review.

The authors use two case studies to point out challenges with Bayesian age-depth modeling – and rightfully demonstrate that the tested models are not without bias and artifacts. I particularly like – and agree with - the repeated call for applying common sense and looking at data with an experienced eye of a geochronologist even when applying models, e.g. as (quotes from your submission follow) 'As frustrating as it may be, in our view none of the tested models can tell us anything better than the actual data themselves', and as 'when testing any chronological model, it is of utmost importance to compare the model outcome with the input data.'. I fully agree and find this an important lesson: look at data, know possible issues – and then think if a model may help and/or is of any help.

Thanks again!

The final statement 'Our study shows that this goal [make use of prior observations to refine the precision, accuracy and robustness] is difficult to reach and that using models to correct measurements appears to be dangerous.'. Well – that really depends on the case and individual data structure in my opinion, and such a general statement should be at least softened when based on two datsets only (why is no reference to the often used models BChron & Bacon made?), and few datasets which are indeed challenging.

Agreed – we have changed our wording, by specifying that according to our case studies, it is high-resolution datasets that push existing models to their limits (more below on BChron and Bacon).

With this I come to my main criticism of this manuscript: the arbitrary selection of models, seemingly influenced by previous work of the authors. When speaking of luminence modeling I ask you to refer to ADMin (https://www.sciencedirect.com/science/article/pii/S187110141730047X) – probably

the model least affected by the spread effect (?), but at the same time slow/unsuitable for large (and these?) datasets.

We have read the article suggested by the referee (Zeeden et al., 2018); however, we find the ADMin model slightly difficult to describe, since it is not defined by any equation. In addition, the nature of errors (systematic VS random) is ignored while the total uncertainty budget is fixed, which seems contradictory. Calibration errors are not random, by definition; so, when in Figures 2 and 6A the model predicts 100% of uncertainty as random, it makes no sense from a physics point of view (there has to be some systematic error). Moreover, the probability density is increasing with the fraction of random uncertainty, suggesting that this function is not integrable – thus, such a function cannot be considered as a probability density. Finally, about the spread effect: it is present in the ADMin model, by definition (since this model only accepts sequences of ages strictly in order); it is indeed visible on the ages presented in Fig. 4 of Zeeden et al. (2018).

Generally, I disagree with the BChron and Bacon models not even being mentioned, as these are really often used.

The reason for not mentioning BChron and Bacon is that our article deals with age estimation under ordering constraints, not with age-depth modelling. These two issues are quite different and, from a modelling perspective, independent questions: in the first case, an age is estimated for each measured sample from a suite, whereas in the latter case an age is estimated at any given depth in a profile – in light of the measured samples. Notably, in our second case study (as is the general case when working on archaeological sites), depth is not relevant – simply because stratigraphy does not follow depth/altitude.

Further comments.

References to *Ramsey* should in my opinion be to *Bronk Ramsey*

Agreed. We found the two occurrences in the literature, but after checking papers by the mentioned author himself, we now refer to Bronk Ramsey.

Line 84: 'event model of Lanos and Philippe (2018)' – could you please introduce this one – it is less known than the one by Bronk Ramsey which you introduce in detail

Agreed.

Line 115: please explain 'Theta matrix'

Agreed.

160ff: Can BayLum model 14C ages!? - that would be different than luminescence modeling, because here 'only' the 14C age is used?

Yes indeed, BayLum does allow modelling 14C ages – as detailed in Philippe et al. (2019).

In Fig. 3 (and others) please include original ages. The problem is that Fig. 3, like Fig. 4, are outputs generated by software (Chronomodel for Fig. 3 and BayLum for Fig. 4); so, unfortunately, we cannot change these figures.

Generally, I find your figures would benefit from clearer explanation in captions, and systematically placing units on axes - ideally all would be on the same age (ka or BC ,please dont mix here).

Agreed.

Abscissa of Fig. 3 : space before bracket missing

Agreed.

Figure 5 and its explanation: ordinate unclear. Why was this only done for BayLum?

Agreed – we have reformulated the caption of Fig. 5 and added more explicit explanations in the text. About why this was only done with BayLum: because the comparison on OSL data with other models would be blurred by the fact that only BayLum starts from aw measurements of OSL data and includes shared errors across OSL samples. In other words, only BayLum takes the specificities of OSL ages into account, whereas all other models treat OSL ages as Gaussian probability densities (with only random errors). So, while the comparison between OxCal, Chronomodel and BayLum is justified for radiocarbon datasets, it is not the case for OSL. We stated this in our introduction (l. 97-100): 'To compare chronological models when confronted with high-resolution datasets, we turn to radiocarbon dating. Indeed, in BayLum OSL measurements are combined in a hierarchical model linking regenerative doses, individual equivalent dose estimation, the central dose parameter of interest, etc. Conversely, OSL ages can be included in OxCal and in Chronomodel, but only in the form of Gaussian probability densities (in practice, an age and its uncertainty). *In particular, unlike in OxCal and Chronomodel, in BayLum it is possible to model shared errors across OSL samples arising from, e.g., equipment calibration errors*.' Note: we added the sentence in italic to make our point clearer in the text.

Line 278: please explain the phase structure here. We are not sure what the referee is suggesting here, since we reproduced the model of Bayliss et al. (2015) and explicitly refer the reader to this original publication.

284: 'between samples OxA-9893 and OxA-23251' – please mark in Figure so that these can easily be found. Agreed, we find that this is an excellent suggestion and we hope it will help readers recognise similar features in their own studies. We actually added two brackets, in black, to highlight the (in our view) two clearly visible concentration effects: one concentrating ages towards younger periods (bottom bracket, between samples OxA-9893 and OxA-23251) and one towards older ages (between samples OxA-9776 and UCIAMS-103138). We added a sentence corresponding to the second sample set in the main text.

286f: I disagree with your statement 'These two bottom-most samples are PL-980252A, whose age lies outside the calibrated age of all samples above' – the densities do overlap. Agreed, there is indeed a tiny probability that sample PL-980252A has an age consistent with the samples above. We added 'almost entirely' in the sentence.

Chapter 3.2.2., and Fig. 8 limited to the lower 17 samples – was the model run for all or these samples? The model was run for all samples, but the figure only shows the 17 bottom-most sample; otherwise, the figure is very difficult to read. We are now more explicit in the text.

Line ~322: please highlight where the spread effect is pronounced why

Same as above regarding the suggestion to highlight the concentration effects, we thank the referee for this very helpful comment. We added another bracket in blue to highlight he spread effect visible in the top part of the sequence. We also added text to describe our observation: prior to modelling the ages stratigraphy, all calibrated ages are essentially the same; yet, in the modelling output, the ages are inconsistent because they are spread apart.

Fig. 10: units on both axes missing – please also include original dating - either as distribution or mean ages. We tried this suggestion, but reading the figure becomes very difficult because of the large number of samples. We believe that adding another dataset is only useful with a smaller number of ages (Fig. 11). Finally, we added units (years) on the x-axis – the y-axis does not have any unit.

352ff: given that Chronomodel and OxCal partly do not overlap the praising of larger uncertainty alone seems unjustified. Here we disagree, because uncertainties estimated with OxCal appear very small (too small in our view) compared to measurement uncertainties. Since the two models disagree, we feel like it is OK to write that 'Chronomodel appears to lie on the cautious side of things' and that this is perhaps an advantage.

In chapter 4.1. I find a prominent feature missing: The duration of the sequence when using OxCal is much shorter than when using BayLum or Chronomodel. This is worrying in my opinion, and the OxCal results seem much more similar to original

ages than the BayLum and Chronomodel results. Especially the outer model ends seem unrealtistic long in BayLum and Chronomodel. The spread effect of the whole sequences seems therefore best captured by OxCal.

Agreed – to some extent. Indeed, BayLum and Chronomodel are strongly affected by the spread effect – while in OxCal it is balanced by a concentration effect. Even if we have not quantitively estimated the duration of the archaeological sequence of Çatalhöyük, we agree with the referee that BayLum and Chronomodel will overestimate this duration. This being said, OxCal will underestimate it (since it underestimates the start of the mound). Overall, it is difficult to say whether OxCal or BayLum best captures the duration of the whole sequence – both models are presumably in error, in two opposite directions.

In line 415 I suggest reference to

https://www.sciencedirect.com/science/article/pii/S0277379103003160

https://journals.sagepub.com/doi/full/10.1177/0959683616675939

As stated above, we are not dealing here with age-depth modelling – so we feel like referring to these two articles is no justified.

It is really good to see the computer code in Supplements. Yet I am wondering why this is only the case for one of the two examples. Further, R code would benefit from better documentation, please do so that also non-R-familiar colleagues can follow what is done why.

Agreed in principle; but the OSL data is not ours, so we cannot publish it. Hence, it will not be possible for readers to reproduce the calculations on the samples used in this study.

Further, I would like you to provide results (data plotted in Figures) in Supplements. We are not sure which data the referee is mentioning here. Which data? In our view, the ten figures already included in the article suffice; and all data are provided in tables for the reader to produce additional figures if deemed necessary.

I am aware of issues with suggesting literature in the review process, and I am asking the editors to have a critical look at these – yet I ask you to consider including the information contained within the suggested literature in your manuscript.

**References**

Zeeden, C., Dietze, M., & Kreutzer, S. (2018). Discriminating luminescence age uncertainty composition for a robust Bayesian modelling. *Quaternary Geochronology*, 43, 30-39.

---

## Author Comment (AC2)

**Review of**

The conflict between sampling resolution and stratigraphic constraints from a Bayesian perspective: OSL and radiocarbon case studies

**28th June 2025**

**Overview**

This paper presents two worked case study examples that show three different modelling approaches can lead to strikingly different results when applied to the same data.

I think this is an interesting result, and important for readers who might be using these models to understand, but I do feel that the current exposition is really quite unclear – in terms of sufficiently describing what the different models do; their interpretation; and the potential reasons for any differences in the resulting inference.

They argue that the differences they see between models are due to the way that they handle stratigraphic ordering, but it is not entirely clear to me that it is solely this – as they also seem to argue that the models implement this stratigraphic information in very similar ways.

If that is the case, then I would expect similar results across the models. Instead it suggests to me that either they actually handle stratigraphy quite differently, or that there are more fundamental differences between the three models (which are not explained) that effectively lead to quite different modelling assumptions; or that (some of) the models have perhaps not converged correctly.

First, we would like to thank the referee for this feedback. In our view, the source of difference between results obtained with the three compared models is clear: despite using the same model for stratigraphic constraints, OxCal implements phases and Chronomodel uses the event model – while BayLum implements no such extra model. This statement appears in the following form in our conclusion: 'Statistical artefacts that should be given special attention are the spread effect (common to OxCal, Chronomodel and BayLum), the concentration effect when using the phase model implemented in OxCal, and unwanted loss of precision when using the event model implemented in Chronomodel.' We acknowledge however that our demonstration must be improved, since it did not convince the referee.

Specifically, they compare:

1. BayLum when modelling OSL dates

2. OxCal, BayLum and ChronoModel when used to analyse a selection of 14C dates from Catalhoyuk

As I said, I think the overall manuscript has highly useful and valuable content for the community and I would recommend publication, but IMO the overall narrative and level of clear explanation really does need to be improved if it is to be of substantial use to the relevant community.

**Major Comments**

**Explanation of the Models**

There needs to be much more, and understandable, detail on the mechanics of the three actual models. Currently, there is quite a lot of written text (which I feel is hard to parse and repetitive in places) but I was not particularly clear on how any of the three approaches actually modelled the data – in particular BayLum or ChronoModel as these are less well-known to me.

I do not think that in this instance, where you are arguing the models are inconsistent with one another, it is sufficient just to refer to the original papers. One needs to understand why the models might be presenting different results to one another (and that must either be because they have different underlying statistical modelling approaches, or that the MCMC they all seem to use hasn't converged properly (which is a serious issue with using MCMC on an ordered parameter space as it becomes highly multimodal).

I understand you cannot describe the entire approach for each, but what are the actual mathematical/statistical models for each that are fitted (what is the likelihood, what is the prior, …). In my view this should be done through short mathematical equations with a suitable notation, not long passages of text which aren't sufficiently explicit.

If it is as simple as just some phase models and stratigraphy I would hope/assume that this can be provided in a couple of general equation (see e.g., Nicholls and Jones, 2001, for a complete and general explanation of the stratigraphic phase model as I presume is fitted in OxCal to Catalhoyuk).

Agreed. We have added equations in our text to make our text more explicit. This being said, it is not clear to us what exactly is implemented in OxCal, all the more since OxCal is not open source.

**Comparison with Nicholls and Jones (2001)**

It is not clear to me that the authors have entirely understood this paper as they seem to argue against the point I see it as trying to make several times (e.g., line 51) while then citing it as an example of an approach to address the problem of under spreading as they see it over. Importantly, I do not see this paper as making a particular statement about using a uniform phase within a boundary. It is about a different aspect – the prior on phase boundaries.

There seems to be a misunderstanding here. We assume that the referee is saying that we are arguing against the modelling approach of Nicholls and Jones (2001) (despite the first sentence structure that we do not follow). That is not the case. We read again our text carefully, paying special attention to places where we quote Nicholls and Jones (2001) and we do not see where the referee gets the impression that we argue against their prior.

My reading/understanding of Nicholls and Jones (2001) is that their main thrust is to argue that if you do not use a sensible prior then you get erroneous overestimation of the spread of stratigraphic calendar ages (i.e., dates that are too far spread). Hence they are arguing precisely that you should compress/shrink the calendar ages. We agree with this reading of Nicholls and Jones (2001). This paper evidently needs to be referenced but I think in a very different way.

Nicholls and Jones' specific, recommended, prior actively aims to penalise the spread of phase boundaries. This problem of over-spreading is well known and theoretically evidenced (e.g., by Stein, 1956). Nicholls and Jones demonstrate this in the context of 14C calibration convincingly using Bayes factors. They also provide a discussion of what underlying depositional model leads to their chosen prior which penalises the spread, and an argument it is more sensible than the model which generates a constant prior density. They are therefore arguing against the idea of this paper(i.e., they are making a case things should be compressed!).

To our understanding, Nicholls and Jones (2001) argue for the use of a uniform prior density on the time span of a phase, to avoid overestimating the duration of a phase. We do concur with them – BayLum and Chronomodel, which do not use the Nicholls and Jones prior, indeed tend to overestimate phase durations. In OxCal, it is not clear to us what exactly happens when phases are encapsulated. It is possible that the behavior of phases in the OxCal model is influenced not only by the choice of prior

distribution, but also by the intrinsic structure of the model itself. The phase model implemented in OxCal involves introducing additional parameters, such as the start and end of the phase (note that there is only one parameter between successive phases in Nicholls and Jones (2001), whereas in OxCal each phase has one start and one end parameter). Consequently, the OxCal model involves a higher-dimensional parameter space – which can impact date estimations and apparently leads to age concentration.

Nicholls and Jones (2001) consider a model that consists of ordered phase layers (with adjoining boundaries $\psi 0, \ldots, \psi M$) and within each layer they assume there are a set of samples with calendar ages. They assume no ordering within the samples inside a layer. This seems to be precisely the model you fit to Catalhoyuk.

Agreed.

Specifically, they assume that the $Ni$ 14C samples within a specific layer/phase $i$ (i.e., the samples between calendar ages $\psi i-1$ and $\psi i$) have calendar age $\theta i,1, \ldots, \theta i,Ni$ and are uniformly distributed in calendar time through the layer (i.e., there is no information, or prior, placed, on their relative ordering). The ordering is really only placed on the phase boundaries, i.e., $\psi i > \psi i-1$

Agreed.

Throughout their paper, they propose a completely consistent prior on the individual sample calendar ages $\theta i,j$ (conditional on the phase boundaries). This is a uniform prior because they don't assume any ordering of the samples within the layer. Their argument is entirely concerned with what prior you should place on the phase boundaries. They propose two – a constant prior that does not penalise the overall range/span of all the layers which they argue is actually highly informative on the overall span and leads to overspread inference; and one that does, i.e., operate to reduce the overall span $\psi M - \psi 0$ which they argue (with theoretical support) works better.

Agreed.

Which it the bit you are arguing against? The uniform phase component of the model within each layer? Or the prior on the spacings of the phase boundaries. These are quite different things.

We are not arguing against the prior proposed by Nicholls and Jones (2001). We observe, however, that OxCal simultaneously concentrates ages (reducing phase duration) and spreads successive ages apart.

As far as I am aware, Nicholls and Jones (2001) do not discuss, or argue for/against the uniform prior on the samples within each layer – this is taken as an assumption of both their priors/models.

If I am correct then I think this therefore means that some needs some significant rewordings are needed in the paper. There is stuff about uniform phase models being needed to reduce erroneous over-spread (where the samples do arise from such a model) but it I'm not sure this is Jones and Nicholls (2001). We are prepared to rephrase our text, but we have trouble understanding what exactly the referee is suggesting.

**Assessment of Model Appropriateness**

I entirely agree with the authors that all users should not treat statistical models as black-boxes, but rather ensure/investigate that the assumptions made within those models are valid for their example/analysis. Also that there is no such thing as a non-informative prior (and that the choice of

prior can have large consequences on inference). This is something which is far too frequently overlooked, especially in Bayesian analyses (which will always give you an answer even if it doesn't make sense).

However, I am a little concerned by the idea/conclusion which I feel goes too far in suggesting that you should perhaps ditch modelling altogether, and that you can conclude the models are wrong primarily because they don't overlap with selected independently-calibrated 14C dates. There is a large literature (backed up by theory) which tells you that independent estimation of any random variables is not a statistically-valid approach and leads to overly-spread estimation (going back to Stein, 1956). We should not encourage people to go back to that.

We see ourselves as modellers and we do not see our article as an invitation to ditch modelling altogether. Rather, our aim is to highlight an (in our view) important modelling problem: that of dealing with stratigraphic constraints. Our impression is that there is currently no satisfying mean (i.e., well-suited software) to deal with this common situation in geochronology, and that users should be aware of this issue.

Of course, one should be concerned by the differences between the inference provided by these models but I think the reasons are more nuanced than you present. As you say none of the models are likely correct – but then neither is independent calibration of each sample. In principle we agree – but we have not identified major problems with independent calibration of each sample. Presumably some of the models fundamentally have components which you subjectively might not support in their construction/prior. I feel you need to draw that out (which would be easier if the actual statistical modelling equations were explicitly laid out). We have now added the equations, as mentioned above.

**Technical/Minor Comments**

Equation 2 seems wrong to me. Should the numerator on the RHS be 1; and the product moved outside of the fraction? Currently the RHS numerator repeats the LHS – which is circular. Equation 2 is not in error, but can be made more explicit as follows. Let us consider a phase bounded by $t_\alpha$ and $t_\beta$. If there are $n$ samples with ages $t_i$, with $i=1$ to $n$, then:

$$\pi\big(t_\alpha; t_i; t_\beta\big) = \frac{\prod_{i=1}^{n} \mathbb{1}_{t_\alpha \leq t_i \leq t_\beta}}{\big(t_\beta - t_\alpha\big)^n}$$

We have included this equation in our manuscript.

BayLum uses IntCal13 – I would suggest that you need to show/state that this does not lead to the differences in inference. It shouldn't as IntCal13 and IntCal20 are very similar in the period of Catalhoyuk but you need to explain/show this (e.g., by providing plot of the two IntCal curves alongside one another in this interval). BayLum uses intCal20; in addition, ages calibrated independently are shown in Fig. 2.

I commend the authors on making their code available but it would be nice if it was on Zenodo/Github rather than simply text in an Appendix.

We find it essential hat our code is available, but we do not feel like it deserves a DOI (so, Zenodo does not seem appropriate); in addition, when it comes to long-term preservation, we feel like an appendix in the journal is better suited than Github where things can disappear, e.g., if an account is deleted or no longer maintained; or if the file is inadvertently modified. Finally, we can send our codes upon request.